# Deep Convolutional Neural Network Regularization for Alcoholism Detection Using EEG Signals

**DOI:** 10.3390/s21165456

**Published:** 2021-08-13

**Authors:** Hamid Mukhtar, Saeed Mian Qaisar, Atef Zaguia

**Affiliations:** 1Department of Computer Science, College of Computers and Information Technology, Taif University, P.O. Box 11099, Taif 21944, Saudi Arabia; zaguia.atef@tu.edu.sa; 2Electrical and Computer Engineering Department, College of Engineering, Effat University, Jeddah 22332, Saudi Arabia; sqaisar@effatuniversity.edu.sa; 3Communication and Signal Processing Lab, Energy and Technology Research Centre, Effat University, Jeddah 22332, Saudi Arabia

**Keywords:** classification, optimization, batch normalization, kernel regularization, convolution, pooling, dropout layer, learning rate

## Abstract

Alcoholism is attributed to regular or excessive drinking of alcohol and leads to the disturbance of the neuronal system in the human brain. This results in certain malfunctioning of neurons that can be detected by an electroencephalogram (EEG) using several electrodes on a human skull at appropriate positions. It is of great interest to be able to classify an EEG activity as that of a normal person or an alcoholic person using data from the minimum possible electrodes (or channels). Due to the complex nature of EEG signals, accurate classification of alcoholism using only a small dataset is a challenging task. Artificial neural networks, specifically convolutional neural networks (CNNs), provide efficient and accurate results in various pattern-based classification problems. In this work, we apply CNN on raw EEG data and demonstrate how we achieved 98% average accuracy by optimizing a baseline CNN model and outperforming its results in a range of performance evaluation metrics on the University of California at Irvine Machine Learning (UCI-ML) EEG dataset. This article explains the stepwise improvement of the baseline model using the dropout, batch normalization, and kernel regularization techniques and provides a comparison of the two models that can be beneficial for aspiring practitioners who aim to develop similar classification models in CNN. A performance comparison is also provided with other approaches using the same dataset.

## 1. Introduction

One of the most prevalent psychiatric disorders and addiction difficulties is alcohol consumption disorder. In this situation, a person is unable to control his or her urge to use alcohol, even though it has negative health consequences such as liver failure, cardiovascular disease, neurological damage, and some categories of cancer [1]. It has the potential to cause significant harm and even death. More than 3 million people die each year because of alcoholism across the world. According to the World Health Organization (WHO), it accounts for more than 5% of all fatalities globally [2]. As a result, it is one of the leading causes of mortality and disability [3]. It also leads to family disintegration and broader societal dysfunction by causing additional damages such as traffic and other accidents, domestic and public violence, and criminality [4].

An early diagnosis of alcoholism will aid individual subjects in becoming aware of their situation and preventing permanent damage. Traditional techniques of measuring the effect of alcohol on a specific person, such as questionnaires, blood tests, and physiological tests, are used in various ways to identify alcoholism. Each approach has advantages and disadvantages. Because of personal and social reasons [5], the questionnaire-based analysis is less accurate. The blood test is not only invasive and unpleasant, but also often unreliable [4]. In this scenario, the electroencephalogram (EEG)-based noninvasive approaches have been presented for alcoholism identification [6,7,8].

The EEG is a noninvasive diagnostic technique that can be used to examine the brain’s neuronal activities. It measures the potentials that replicate the electrical activity of the human brain. EEG signals are a good source of obtaining information about the neurological conditions of a person [9]. Thanks to its low cost and noninvasiveness, automatic EEG signal classification is an important step towards their use in practical applications and less reliance on trained professionals [10]. This approach is based on the hypothesis that brain signals fluctuate as a function of the brain’s functioning condition. Multiple electrodes are placed in various locations on the subject’s scalp during the EEG test to capture the electrical activity of the brain [11]. The electrodes are positioned following norms [12]. The EEG pulses are converted into signals by each electrode, which are then amplified and conditioned via a bandlimited filter [13]. A similar analog processing chain is implemented for each EEG channel. It prepares the EEG signal for proper digitization. Next, the processed signal from each channel is digitized by using appropriate analog-to-digital converters (ADCs). The digitized signals are conveyed to an electronic EEG recording machine, where they are converted into waves and analyzed on a computer screen. EEG is widely used in cognitive psychology, neuroscience, and neurophysiological research for a variety of reasons.

### 1.1. Approaches in EEG Signal Analysis

Existing work in EEG classification can be broken down into several streams. One stream of approaches is focused on the removal of various kinds of noises and other artifacts added to the main signals from physiological, environmental, and technical sources as part of acquisition and processing [14]. These include the works [15,16,17,18]. The second stream of efforts is aimed at the pertinent feature extraction from the preprocessed EEG signals. More recent efforts have combined the mined features with the machine learning classifiers for the classification of signals. In this context, robust machine learning algorithms, namely K-nearest neighbor (KNN), support vector machine (SVM), decision trees, logistic regression, least-squares support vector machine (LS-SVM), random forest (RF), rotation forest (ROF), and bagging have been used [6,7,8,11,13]. Considering the stages involved from preprocessing to feature extraction and classification, some approaches use domain-specific processing pipelines to streamline the development process [19]. However, these methods further reduce the flexibility and generalization of the approaches.

In the last few years, deep neural networks (DNNs) have evolved to gain influence in various classification tasks. DNNs are known for automatic feature extraction [20], and due to their ability to classify nonlinear relations between the input and output, they have been a powerful tool for EEG classification in recent years. It has been shown that features extracted through convolutional neural networks (CNNs) have more representative potential than the traditional feature extraction methods [21]. Another popular concept in DNNs is transfer learning. In transfer learning, a machine learning model learned for one task is reused in another similar task. Thanks to the availability of several successful deep learning models in the domain of image analysis and computer vision, transfer learning in EEG signals involves converting a one-dimensional EEG signal into a two-dimensional image and applying an existing portion from a learned model from the image domain for classification of the two-dimensional EEG images [22,23]. Gong et al. [24] provide a detailed description of the evolution of EEG signal classification from simple statistical methods to deep learning in the last decade.

In a review of the studies for classification of EEG signals for a wide range of tasks, by Craik et al. [10], it was found that the average accuracy of deep learning methods that used signal values as input performed better than the methods that used images or calculated features as input. This was a motivating factor for us to start our approach with the raw EEG signals as input. The review also mentioned that no previous study had analyzed if “deep learning can achieve comparable results without any artifact cleaning or removal process”. Comparatively, as a novel contribution, the current approach does not perform any artifact cleaning and achieves high performance on various evaluation metrics.

While there have been a variety of tasks related to EEG signal analysis, the identification, analysis, and treatment of alcoholism have been given less attention by researchers. Several recent reviews in the last couple of years [10,19,24,25] covering the state of the art in EEG classification of brain-related activities have identified about a dozen domains of application. However, it seems that more efforts have been spent in EEG analyses in the other domains of neuroscience and neuronal activity classification as compared to alcoholism. Only one [25] of these studies has reported alcoholism classification using EEG, and none of the cited studies have applied any form of deep learning for classification. In this work, we are particularly interested in the analysis and classification of EEG signals for alcoholism detection. From a set of EEG signals, we should be able to identify for each signal if it belongs to a normal person or a person with alcoholic predisposition.

### 1.2. Objectives and Research Contribution

The objective of this study was to investigate a simple, but effective, deep learning model—comprising CNN architecture—for the analysis of brain EEG signals for binary classification of alcoholic and nonalcoholic persons. The novelty of our approach is that we perform our analysis on only 2 of the 61 available channels in the EEG dataset. We do not carry out any preprocessing of the signals. Unlike existing studies, which focus on the manual tasks of noise removal and feature extraction from EEG signals, in the current work we give raw, normalized EEG signals to a CNN for automatic feature selection in the temporal domain of the signals. To deal with the scarcity of data, data segmentation is performed without adding any artificially generated data. Furthermore, the neural network architecture utilized is very compact and consists of only four convolutional layers with associated pooling and normalization operations. The number of parameters used by our classification model is also very small, ensuring efficient data processing. The article not only compares the optimal model with a baseline model but also compares with existing approaches that employ neural networks for EEG classification. Because the majority of the papers are hard or impossible to reproduce [19], we make available our code and explain in detail the stages in the development of the complete neural network architecture so that our work can be reproduced by other interested researchers. We also explain various optimization tasks to improve the accuracy of our model in the hope that the research community can benefit from applying similar techniques with better knowledge in their experiments.

The remainder of the article is organized as follows: Section 2 details the background on EEG signal-based classification and identifies some relevant work as well as a comparison of some approaches with ours. Section 3 is about the materials and methods used in this study. Section 4 describes a baseline CNN architecture for classification followed by the description of methods and their results for improving the baseline architecture. In Section 5 we present, and in Section 6 we discuss and evaluate the results. Section 7 concludes the article with future work directions.

## 2. Background and Related Work

Alcoholism is an important societal issue, and there have been only a few studies dealing with the analysis of EEG for alcoholism detection. The absence of alcoholism-related studies in these reviews may indicate a lack of coordinated efforts, but we think it is mainly because of the availability of limited data for this task. When compared to tasks of EEG analysis such as a brain–computer interface or sleep analysis, the researchers have a significant choice in terms of the number of subjects, age and gender of the subjects, duration of the recordings, etc. [19]. In the case of alcoholism, however, the UCI-ML (formerly UCI-KDD) dataset (explained later) is the only mainstream publicly available dataset that has been used in dozens of studies. A possible reason for the lack of an alcoholism dataset may be the stricter conditions in the preparation and curation of the dataset. For example, the UCI-ML alcoholism EEG dataset required both the alcoholic and nonalcoholic participants to have abstained from drinking alcohol in the past 30 days. Finding subjects that meet such a criterion is a difficult task. Moreover, people may not feel comfortable when they are being analyzed for alcoholism due to certain barriers and social stigmas such as embarrassment, fear of losing the job, or concerns about what others might think about them [5]. Despite these issues, analysis and treatment of alcoholism is an important issue for an individual as well as for society and any effort in this regard will bring us one step closer to better analysis and treatment of this disorder. The use of EEG is one important tool in advancing the research in this direction rapidly.

We can divide the existing approaches for EEG-based classification into those that use feature extraction with traditional machine learning classifiers and those that use deep learning methods.

### 2.1. Feature Extraction and Machine Learning

Researchers have proposed several traditional techniques for feature extraction from EEG signals, such as time domain, frequency domain, time–frequency analyses, wavelet analyses, entropy analyses, and energy distribution [9], or the combination of two or more of such methods [26]. A problem with feature extraction is that it is not only computation-costly, but also laborious and time-consuming. Furthermore, as manual data processing is highly subjective, it is unlikely that other researchers may reproduce the results. Despite these limitations, more than a quarter of the studies reviewed by [10] performed manual removal of artifacts.

More recently, the traditional techniques have been augmented by the application of machine learning for signal classification. These methods depend on engineering techniques such as EEG rhythm feature, analytic wavelet transform, functional connectivity, graph and spectral entropies, and empirical mode decomposition (EMD) [6,27,28]. Jiajie et al. [29] used approximate entropy (AE) and sample entropy (SE) as feature extractors and SVM, KNN, and ensembled bagged tree as classification methods in the development of a clinical decision support system for alcoholism classification. With this approach, they could achieve sensitivity and accuracy of up to 95% on the UCI-ML dataset with quadratic SVM. The authors of [8] provide a summary of some methods of feature extraction with different classifiers and a comparison of their performance on the UCI-ML dataset.

Automated techniques such as independent component analysis (ICA) for artifact removal, principal component analysis (PCA), and local Fisher’s discriminant analysis (LFDA) are some of the methods that are applied as preprocessing before any classification methods are applied to the EEG signals [30]. Ren and Han [31] combined linear methods (autoregression, wavelet transform, and wavelet packet decomposition) with a nonlinear feature extraction method (sample entropy) on the EEG signal and then removed the redundant features using class separability methods. The classification was then performed using ensemble extreme learning machines based on linear discriminant analysis (LDA). Rahman et al. [32] have shown that ICA performed better in the instance-based learning method, KNN, while PCA had better results when used with a deep learning (bidirectional long short-term memory) model. Thus, one must carefully choose a feature selection method with the type of classification method adapted. Finally, image features of the signals, such as Fourier feature maps [33] or 3D grids [34], are some feature-based methods. Saminu et al. [9] provide a summary of techniques that combine traditional feature extraction methods with machine learning classifiers for EEG signal classification.

Bavkar et al. [35] used the gamma-band power as a feature in EEG signal on the entire dataset. They compared a total of 13 machine learning classifiers such as linear discriminant, linear SVM, and quadratic SVM. With 61 channels, and using the ensemble subspace KNN classifier, they obtained the maximum accuracy of 95.1%. To carry out a classification using a reduced number of channels, they applied various optimization algorithms such as genetic algorithm and binary gravitational search algorithm (BGSA), but none of the optimization methods could give an accuracy of more than 94%.

Feature extraction is a time-consuming task and requires specific knowledge of the domain and expertise [24]. Moreover, when using traditional machine learning methods, one must experiment with a significant subset of machine learning classifiers before being able to identify the important features and the best-performing classifier. To counter these limitations, deep learning-based approaches have emerged as a way out. Deep learning does not require manual feature selection or extraction, but important features are learned by the deep learning model automatically.

### 2.2. Deep Learning-Based Approaches

DNNs are a powerful tool for the classification of complex nonlinear systems. Of the various deep learning architectures, CNNs have been found to be the most suitable ones in situations such as the analysis of EEG signals; 40% of the deep neural architectures are CNNs, which have been the architecture type of choice since 2015 [19]. We identify some approaches that have used CNNs for the classification of EEG signals. Chaabene et al. applied CNN for drowsiness detection [36]. For EEG signals, they chose 14 channels followed by preprocessing for noise removal and band annotations in the refined signals. They also used data augmentation to artificially create new training instances from the previous ones. By utilizing four convolution layers, one max-pooling layer, and two fully connected layers, they had more than 14 million parameters in the network to classify a person either in the drowsy or awake state. Without network optimization, they achieved the best test accuracy of 79%. After doing some optimization in their network architecture, by adding various normalization and dropout layers, their test accuracy improved to a maximum of 90% with 7 channels.

Qazi et al. [37] applied a multichannel pyramidal convolutional neural network (MP-CNN) for EEG signal classification of alcoholism. They started with a total of 61 channels from five different brain regions and by assessing the performance of each channel one by one. Of the five developed models tested with varying architectures and several parameters, the best model uses 19 best-performing channels as input and gives an accuracy of 100% with 14,066 parameters. A major limitation of their work is the lengthy job of preprocessing the process of trial-and-error involved in the selection of the best channels and evaluation and comparison of five different CNN models.

A slightly different approach was taken by [38], as they developed two new activation functions to speed up and improve the performance of CNN in EEG classification. With one of the activation functions, they could achieve an accuracy of 92.3% on the alcohol EEG dataset, which is an improvement over the usage of the default activation functions of softmax or sigmoid.

In the case of transfer learning, Xu et al. [23] used the VGG-16 CNN model, originally developed for the general image classification task, for the classification of motor imagery (MI) EEG signals. The newer model consists of the same initial layers as used in VGG-16, except for the final output layer, which is fine-tuned in the target model using the EEG dataset. The EEG signals are converted in time–frequency spectral images using short-time Fourier transform (STFT) before applying them as input into the target model. The classification is done by applying 2D CNN on these images. In total, their CNN model contains 13 convolutional layers, 5 max-pooling layers, and 2 fully connected layers. The average reported accuracy for all subjects was 74.2%. This is 2.8% better than their designed CNN. Srabonee et al. [22] also achieved 98.13% classification accuracy with the transformation of EEG signals into 2D images. In addition to the transformation of images, they also performed Pearson’s correlation analysis on the images before using them as input into the CNN model.

Transfer learning was also performed by Zhang et al. [21], combining the low- and high-frequency signals from 11 high-variance channels from the UCI-ML dataset. This translates into a visual heatmap of brain activity in different parts of the brain. The authors used a combination of 3 traditional feature extraction techniques (gray-level co-occurrence matrix (GLCM), Hu moments, and local binary patterns (LBP)) and 12 deep learning feature extraction models with seven machine learning classifiers (KNN, SVM linear/polynomial/RBF, RF, MLP, and NB). After experimenting with a total of 105 possible combinations, the model with MobileNet CNN architecture as feature extractor and SVM-RBF as classifier achieved about 95% score in accuracy, precision, and recall metrics on the UCI-ML EEG dataset.

The potency of deep networks is due to their depth and the activation properties of the hidden layers. As an example [39], carried out the classification of alcoholism using EEG signals by five machine learning algorithms and a multilayer perceptron (MLP, a shallow neural network with one hidden layer only). With several feature selection approaches experimented on, the MLP did not perform as well as the rest of the algorithms. It performed worst in most of the cases.

From the above analysis, we can conclude that there is a diversity of methods for the classification of EEG signals, and the competition to find the best method is still ongoing. In our opinion, the best method should be selected based on flexibility, efficiency, and overall performance on various metrics. Unfortunately, many of the mentioned approaches report only a few evaluation criteria such as accuracy, while ignoring other metrics like precision and recall, which can be possibly lower, hence diluting the overall performance. Thus, we intend to report our performance in terms of several metrics in this article.

## 3. Materials and Methods

### 3.1. Experimental Setup

We implemented our model in the Python language using Keras API (https://keras.io/ (accessed on 25 July 2021)) for deep learning with Tensorflow (https://www.tensorflow.org/ (accessed on 25 July 2021)) as the backend. For program development, we used the GPU environment in the Kaggle platform (www.kaggle.com (accessed on 4 August 2021)), which can execute the model several factors faster than a CPU.

### 3.2. EEG Dataset

The dataset for EEG signals was obtained from the University of California at Irvine Machine Learning repository [40] and was produced for research by H. Begleiter [41]. The complete dataset consists of 122 subjects, each with 120 trials with two different stimuli. The subjects were divided into alcoholic and control groups. Each subject was exposed to either a single stimulus or two stimuli in either a matched condition where S1 was identical to S2 or in a nonmatched condition where S1 differed from S2 [42]. After removing trials containing unwanted eye and body movements, the EEG recordings of each class were retrieved.

The EEG signals were recorded using the placement of 64 electrodes on the head according to the International 10/20 system [12]. Frontal polar (FP), frontal (F), temporal (T), central (C), parietal (P), ground (G), and occipital (O) areas are represented by the 10/20 system elements. The outcomes of electrodes are very sensitive to noise. Therefore, each electrode outcome was amplified and then passed through a filter with a pass-band of [0.02 Hz, 50 Hz]. This band-pass filter not only limits the signal bandwidth but also avoids the low-frequency baseline wander noise. In the next step, the data were sampled at a frequency of 256 Hz with an analog-to-digital converter (ADC) of 12-bit resolution.

Experiments were conducted by using a partial dataset in this study since the publicly accessible entire dataset is incomplete, with certain trials containing empty files or tagged as “err.”. For both classes, normal and alcoholic, 60 EEG recordings were kept. For balanced representation, 30 recordings were considered from each category. Following the work presented in [6,43], 61 electrodes were used to acquire the EEG signals and the remaining 3 electrodes were used as reference. The Cz electrode was utilized as a reference, while the X and Y electrodes were used to capture horizontal and vertical bipolar signals, respectively. The baseline filter removed artifacts such as eyes and muscle movements from intended records [19]. Each conditioned recording has 8192 samples and is 32 s long. Each preprocessed record was further divided into four portions. Each portion comprises 2048 samples and is 8 s long. Figure 1a,b shows sample plots from normal and alcoholic persons.

### 3.3. Data Segmentation

To increase the data size, each 8 s length EEG portion was divided into four 2 s length segments, each containing 512 samples. The process of segmentation was carried out by using the rectangular window function, with the windowing operation [17] given by Equations (1) and (2):(1)$$zw_{n}=z_{n}\times w_{n}$$
(2)$$zw_{n}=\overset{\frac{\tau }{2}}{\underset{\frac{-\tau }{2}}{\sum }}z_{n}$$

Here, $z_{n}$ is the digitized version of EEG band-limited signal, obtained from the considered dataset [16]. $z_{wn}$ is its segmented version. $w_{n}$ is the window function coefficient vector. Its length is equal to τ=2.0 s, and it contains 512 coefficients, each of magnitude 1. This operation of windowing breaks the longer EEG signal in smaller segments. Each segment is considered as an instance. In total, 960 instances were studied, out of which 480 belong to the normal class and the other 480 belong to the alcoholic class.

### 3.4. Data Normalization

An important aspect of the current work is that we did not perform any special preprocessing tasks on the raw input data, with the only exception of standard normalization. In general, the signals in the data can have arbitrary positive or negative values (as shown in Figure 1). To reduce any wide dynamic ranges in the signals, it is suggested that data normalization be performed before the training process [44]. The standard normalization process scales the data so that it has a mean (μ) of 0 and a standard deviation (σ) of 1.

### 3.5. CNN for Feature Selection

CNNs are a special kind of artificial neural network for processing data that are usually in a series in 1 dimension, e.g., speech or EEG/ECG signals, or in 2 dimensions, such as images. The principal operation in CNN is that of a *convolution,* which is a specialized kind of linear operation on two functions of a real-valued argument [45]. The first function of the convolution operation is the input, and the second is known as the *kernel*, while the output is known as the *feature map*. Because of the smaller size of the kernel compared to the input, convolution requires fewer parameters due to *sparse connectivity*, which not only reduces the memory requirements of the model but also improves its statistical efficiency [45]. A kernel of the convolution function can extract only one kind of feature at different input locations. To extract various kinds of features, we apply more than one convolution function in a single CNN model. The convolution function is usually followed by a pooling function, which modifies the output of the convolution layer by downsampling. It does so by replacing the output at a certain location with the summary statistics of the nearby outputs using functions such as maximum or averaging functions [45]. The convolution and pooling operations are usually represented by a single convolution block.

CNNs have been used in problems such as speech recognition, image classification, recommender systems, and text classification. More recently, CNNs have been shown to classify EEG brain signals for autism [46], epilepsy [46,47,48,49], seizure detection in children [50], schizophrenia [51], brain–computer interface (BCI) [52], alcoholism predisposition [21,37], drowsiness detection [36,53], and neurodegeneration and physiological aging [54] into normal and pathological groups of young and old people.

#### 3.5.1. Fully Connected Layer for Classification

The convolution block is followed by fully connected or dense classification layers. In general, several such layers may be needed for improved discrimination. Depending on the problem, each layer may have a specific activation function. For binary classification, the sigmoid or the logistic regression activation function is used.

#### 3.5.2. Hyperparameter Tuning

In addition to the parameters for model definition, e.g., the number of layers, their types, and their activation functions in a neural network, a set of hyperparameters also governs the performance of a model by controlling various aspects of the algorithm’s behavior [45]. The hyperparameters can be tuned manually or automatically, and the range of their values can affect the time and running cost of the algorithm. Examples of hyperparameters include the type of optimizer, the learning rate, the input batch size, dropout rate, and the convolution kernel width. These are the hyperparameters as they are usually not learned by the algorithm on the training set, but their values can control the model capacity [45]. The hyperparameter values are adjusted based on the validation set once a model is learned from the training set. The final set of hyperparameters is fixed on the test set after seeing the generalization error. Section 4.3 explains the various forms of hyperparameter tuning in this work, e.g., learning rate, dropout rate, and kernel width.

#### 3.5.3. Performance Metrics and Evaluation

For binary classification, accuracy is the best measure. We mainly performed all the optimizations using accuracy as the main metric. For the final model, we also report the cross-validation results. In addition, the widely used metrics of precision, recall, F1-score, Cohen’s kappa, and area under curve (AUC) were also determined.

## 4. CNN Architecture for EEG Classification

The proposed work applies CNN for the classification of an EEG signal as belonging to an alcoholic or a normal person. The main components of the model are one-dimensional convolutional layers and a dense (fully connected) layer.

### 4.1. 1D-Convolution and Pooling

We started with a relatively small number of layers where each layer had a small kernel of size 3 and the number of filters was 8 to obtain a modest accuracy of about 72%. The capacity of the model was increased by adding more layers and increasing the size of the kernels and the number of filters progressively in various layers. This was done by constructing a different grouping of the same number of filters ranging from 8 and increasing by a multiple of 2 until there were 128 filters in each layer. Similarly, by mixing various other sizes of filters, we kept measuring the change in the error loss and accuracy. The finally selected model had four convolution layers with 16 filters in the first, 32 in the second, and 64 in the last layer. The kernel size was fixed at 15 in all layers. The convolution stride was also fixed to two steps in every layer. Rectified linear unit (ReLU) activation function was used at each convolution layer for bringing nonlinearity in the process. Reducing the value of any of these parameters resulted in decreased performance, while increasing the value did not achieve any performance gains.

This is the general design pattern for CNN—the number of filters is increased in the latter layers, starting with a relatively small number at the start. Moreover, a convolution operation is mostly followed by a spatial pooling operation. Max-pooling is generally a preferred approach over other forms of pooling such as averaging, giving better results. The final convolution operation is also generally followed by a global max-pooling operation.

The convolutional/pooling layers are responsible for feature selection. To classify the EEG signals, a single fully connected layer was applied that uses the sigmoid activation function for binary classification and the binary cross-entropy method for loss minimization. The architecture of the model is shown in Figure 2.

### 4.2. Training and Testing of the Model

After the data segmentation (Section 3.2), we had 960 instances where each instance had 512 dimensions. The training was carried out using mini-batch gradient descent with a batch size of 64. Going through the successive convolution and pooling layers, the dimensions are reduced because of feature selection at each layer as shown in Figure 2.

As the model starts learning, a small validation set is used after each epoch to adjust the weights. We fixed the validation set to be 20% of the size of the dataset; the model automatically chooses validation examples in each iteration. At test time, the model accepts input examples in the same dimension as the training data.

Figure 3a,b shows the loss and accuracy for the developed model. As can be observed, the training loss and accuracy continued to improve (low training error vs. high accuracy), but the validation loss and accuracy stalled after a few iterations. If the number of iterations was increased further, the training data would achieve an accuracy of 100% without any improvement in the validation accuracy, which remained at 92% in the best case. This is a case when the model overfits the training data. Increasing the capacity of the layers in terms of bigger kernels or adding more filters cannot improve the validation accuracy, as that will only help in memorizing the training data.

Thus, we needed to fine-tune the CNN model to maximize the results on the validation set. Once a desirable validation accuracy is achieved, we can then use the model on the test data to assess its final performance.

### 4.3. Optimizing the Neural Network Model

We considered the previously developed model as a *baseline model* and brought improvement to it to achieve high accuracy on the validation set. To avoid overfitting of the network on training data, a technique called regularization was applied to the model. Regularization adds a penalty to the model so that the model does not overfit the training data. Three regularization techniques were used: dropout, batch normalization, and L1/L2 regularization. Although regularization is an important technique for obtaining better generalization in DNNs [45], more than half of the papers in EEG classification did not mention using any regularization techniques [19]. Since regularization methods helped us in achieving good results, it would be essential to give details of the different regularization techniques we have applied in our approach.

#### 4.3.1. Batch Normalization

The purpose of normalization is “to make different samples seen by a machine-learning model more similar to each other” [55]. Normalization of the input was performed in the preprocessing step. However, during input processing, data are processed by the layers and transformed into a wide range of values that need to be normalized. Batch normalization [56] tries to achieve the same effect in the DNN. In our case, we applied batch normalization to the output of all max-pooling layers. The normalization of layers improved the accuracy by a few percent.

#### 4.3.2. Dropout Layers

Dropout layers were proposed by Srivastana et al. as a simple way to prevent the neural network from overfitting [57]. Dropout consists of dropping out some units along with their connections from the neural network, significantly reducing overfitting. Dropout regularization was used in all the layers with a dropout ratio set to 0.1. An increase in dropout ratio to various other values resulted in a decreased performance.

#### 4.3.3. L1/L2 Regularization

One way to reduce the complexity of the model is to put constraints on the model weights. This is done by adding a cost with the large weight in the loss function. This cost can come in two ways: L1 and L2 regularization. In our case, the training was regularized by applying both L1 and L2 regularizers simultaneously in the fully connected layer. For both, the penalty weight was set to 0.01. Using these regularizers helped in the early convergence of the network and improving the accuracy as well.

#### 4.3.4. Optimizer, Learning Rate, and Early Stopping

After evaluation of various commonly used optimizers, e.g., Adam, Adagrad, and Nadam, the best overall results were provided by the RMSprop optimizer, which we opted to choose for all our experiments. The learning rate is one of the most important hyperparameters [45]. The Keras deep learning library helped decide on the optimal learning rate. Using the built-in *learning rate scheduler* callback function and starting with a learning rate of 1 × 10^−8^, we measured the performance improvement in various training epochs while increasing the learning rate by a small factor to finally reach the value of 0.1. The best performance (in this case the minimum training loss) was achieved with a learning rate of 6 × 10^−2^. Early stopping is a hyperparameter that allows the learning process to stop when there is no improvement in the performance beyond a certain threshold. We configured the training process to be stopped when there was no decrease in the training loss in the last 15 iterations.

## 5. Results

The training was carried out for a total of only 100 epochs (or 1000 iterations considering that each epoch comprised 10 iterations as the input data of size 614 training examples was divided into batches of 64). It should be pointed out our network converged very quickly as compared to many deep learning models where the number of iterations can reach up to several million [58].

Table 1 shows the comparative results of the two models on the test set: the baseline, unregularized model versus the optimized, regularized model for a single run. The regularized model outperforms the baseline model in every aspect, but its value is starkly high for Cohen’s kappa. The kappa value shows the inter-rater agreement. In the case of the baseline, we can see that lower precision resulted in lower kappa, and lower precision was the result of misclassifying a normal person as alcoholic (false positive). In contrast, the regularized CNN model had much better kappa due to high precision and recall.

To evaluate the performance of the final, regularized model more objectively, we carried out K-fold cross-validation [45] using K = {3, 5, 10}. K-fold cross-validation is the preferred approach for model evaluation when the available dataset is not very big. Moreover, we also experimented on varying batch sizes ranging from 2^2^ to 2^8^.

The total data (*n* = 960) were divided into a 20% test set (*n_test_* = 192)*,* while the remaining data were split into training and validation according to the value of K. For K = 3, two-thirds of the data were used for training (*n*_tr_ = 512), while one-third of the data were used for validation (*n_val_* = 256). As the number of folds increased to 5 and 10, the training data were increased (4/5 and 9/10), while the validation data were reduced (1/5 and 1/10).

Table 2 shows the results of K-fold cross-validation and variation in the batch size. The validation columns show the average accuracy of K-runs and their standard deviation for each value of K. The test columns show the accuracy of the test set. As can be seen, the average performance across the folds kept increasing, and the best results on test data were obtained for 10-fold cross-validation. The final column shows the best run of all the folds in the case of 10-fold cross-validation, which achieves an accuracy of as high as 100%. On the dimension of batch size, we see varying results for different folds, without any specific trend, except that larger batch sizes had worsening performance both for validation and test datasets.

We can draw the following conclusion from Table 2: as the number of training samples increases from K = 3 (*n_tr_* = 512) to K = 10 (*n_tr_* = 692), the accuracy also increases. This is in agreement with the general principle of machine learning: increasing the number of samples increases the accuracy of the model [45].

## 6. Discussion

As identified in Section 2, due to the availability of only the UCI-ML EEG dataset for classification of alcoholism in the public domain, almost all the approaches for alcoholism classification [21,29,31,37,39] use this dataset. The dataset can, thus, be used as a benchmark for performance evaluation of various classification approaches. Table 3 provides a comparison of these approaches with ours.

Most of the mentioned approaches achieve an accuracy of more than 95%. However, due to the complexity and generality of the approaches, some may be more promising than others. For example, transfer learning is one successful approach for classification in image processing and computer vision. Zhang et al. achieved above 95% accuracy using transfer learning [21]. However, the application of this approach is not straightforward. The selection of layers to be learned versus those to be reused and the initialization of weights are some fundamental things to be learned from experimenting with the target dataset. Approaches such as [31,36] use deep learning features but are preceded by a pipeline of preprocessing tasks. Similarly, in [37], the authors went through a laborious job of testing various combinations of channels on different architectures of CNN models before finding the one with the best performance. In all these approaches, the dataset-specific model learning implies that the tasks cannot be generalized to other, newer EEG models for alcoholism classification.

Compared to these approaches, generic approaches that use standard CNN models can work well on a variety of tasks, and their generalization capabilities are not affected much on different datasets [45]. The approach used by [22] applies a simple CNN model for EEG signal classification. Their architecture and their performance evaluation results are much like ours except that they performed the preprocessing in the form of Pearson’s correlation and conversion of 1D signals into 2D spectral images. Compared to their preprocessing tasks, we improved our architecture using various regularization methods. We believe that our work is more advantageous because regularization techniques achieve better generalization on previously unseen data and are preferred methods for improving the performance of DNNs [45]. Thus, the current work can be considered as a step towards the advancement of EEG signal classification using generalized DNN models.

Our work also demonstrates that data segmentation can improve the classification task accuracy. As EEG signals are nonstationary, and their statistics vary over time, it is said that a classifier trained on one part of user data might not generalize to some other part of the same person [19]. However, we have seen that segmenting a signal into four parts, in our case, yields impressive results. Thus, classification using different parts of the signal may be feasible in some situations. Some approaches work on outlier detection to improve their performance, e.g., [21]. However, this results in a model that loses the quality of being robust and tolerant in the view of generalization. Thus, we do not apply any outlier detection in our methodology.

Convolutional neural networks (CNNs) have been used in many application areas for classification of audio, video, or text data, while very limited work can be found in the use of CNN in EEG classification. CNNs are efficient in processing inputs of large dimensions thanks to their properties of weight-sharing and sparse connections [45]. These properties not only reduce the number of parameters but also reduce training time and enhance training effectiveness [24]. Data scarcity is one issue when it comes to using deep learning models for medical analysis. On the other hand, training on large datasets means longer times for model fitting and evaluation. The best solutions are those that try to obtain the best results on smaller datasets. As such, most of the existing approaches that used the referenced dataset of UCI-ML worked on the smaller dataset containing only 120 trials of alcoholic and 120 trials of normal subjects’ diagnosis [6,7,8]. In this work, we also employ a total of 240 EEG segments, 120 from alcoholic and 120 from nonalcoholic subjects, without introducing any data through augmentation. Further experiments are needed to see the results if the data are divided into even smaller segments.

Simonyan and Zisserman [58] have shown that very deep CNN architectures with small convolution filters in all layers perform very well in areas such as image recognition and natural language processing. However, in the current work, the use of small convolution filters did not achieve good results. The optimal results were obtained with 1D convolution filters of size 15 in all the convolution layers. This discrepancy may be due to the reason that our architecture is not *very* deep, as mentioned in [58]. Alternatively, it could be that the convolution filter size is reduced at the expense of increasing the depth of the current network. We leave the comparative analysis for future work, as it might involve optimizing other parameters as well.

### 6.1. Importance of Regularization in DNN

Regularization has played an important role in achieving high performance in the classification task. Batch normalization, dropout layers, kernel regularization, and learning rate were the hyperparameters that were tuned in this work. This caused only a very small change in the number of parameters to be learned by the network. Table 4 shows the comparison of the architecture of the baseline CNN model with the regularized model.

### 6.2. Limitations of the Study

One of the limitations of applying deep learning models is that they require a large amount of data for training the network. In case of limited data, a number of data augmentation techniques are applied to generate synthetic data [59,60]. Deep learning methods are also known to require high computation and suffer from slow convergence [24]. However, the availability of advanced processing units such as the graphical processing units (GPU) and the tensor processing units (TPUs) have solved the computation issues to a large extent [61]. To solve the convergence issue, various methods have been proposed recently. This article details the usage of such methods to achieve state-of-the-art performance in EEG signal classification.

It is well known that initialization of network weights is important and bad initialization can result in instability of gradients, thus affecting the learning process. This is considered as a regularization technique [62]. However, in the current work, we did not focus on any weight initialization technique, and random weights were chosen by default by the network using a small Gaussian value with mean 0, so it is quite possible that the model performance could be affected if better/worse initial weights are chosen [63]. Other techniques such as the input perturbation technique can be explored to understand the causal relationship between the input and the decision of the model [19]. We leave this aspect of network optimization for future work.

A weakness of the current work is that for achieving the best model, some fine-tuning of the neural network model was required. However, this tuning of parameters and hyperparameters was limited to only one specific CNN model. In comparison, other similar approaches went through a laborious job of tuning several candidate models before choosing a final one. Zhang et al. [21] experimented on 12 different classifiers, adjusting various hyperparameters of each one before settling on a final model. In total, there were 105 combinations for feature extraction methods and classifiers.

## 7. Conclusions

Detection of alcoholism is an important social issue. EEG is an important tool for the identification of alcoholism. This article explains the use of CNN in the classification of EEG signals for alcoholism identification. Various regularization techniques were explained so that other researchers and practitioners can build on the present knowledge to create more efficient and better-performing CNN architectures. The methods described in the article do not apply to EEG classification only and can be applied in a wide range of applications of CNN.

With the current methodology, we achieved an accuracy of up to 98% on the UCI-ML dataset while also obtaining good results in precision, recall, AUC, etc. Further investigation may be needed in the form of weight initialization, input perturbation, and data segmentation choices, which have been identified as the near future work.

## Figures and Tables

**Figure 1 sensors-21-05456-f001:**
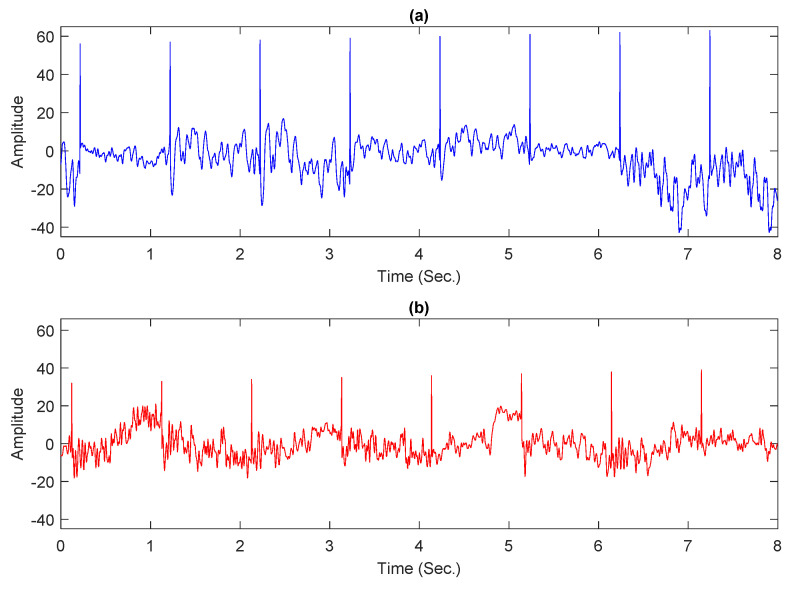
Examples of EEG signals: (**a**) normal person; (**b**) alcoholic person.

**Figure 2 sensors-21-05456-f002:**
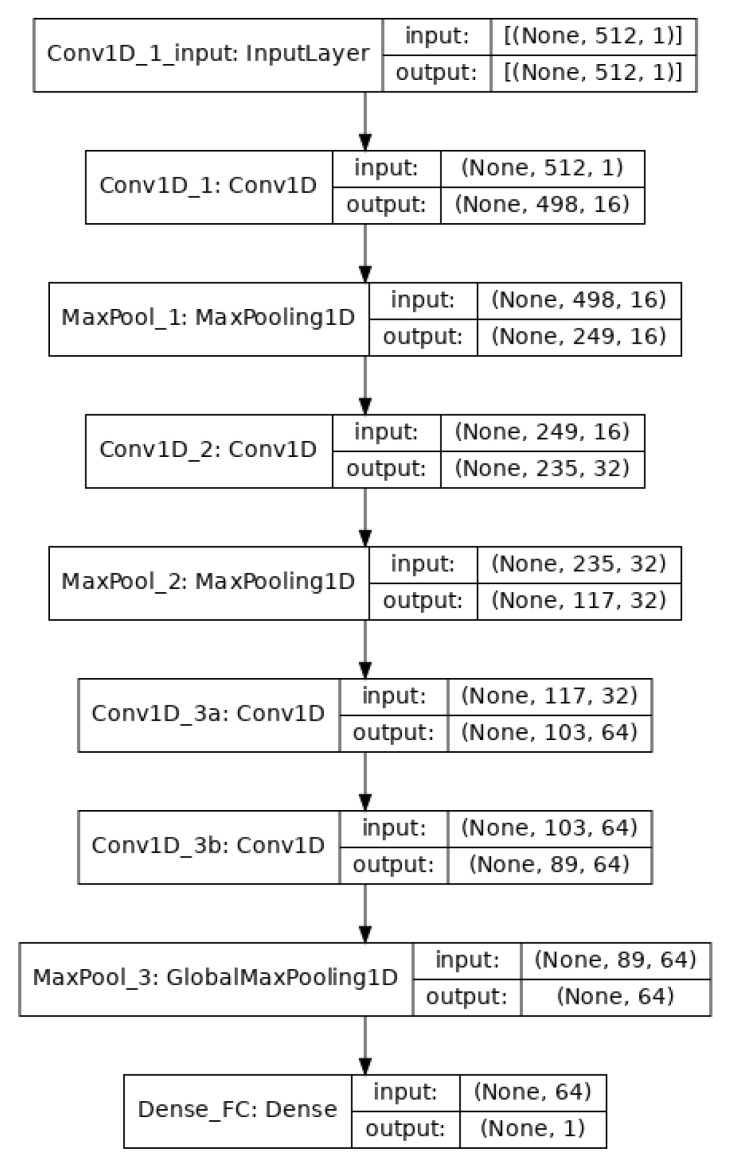
The architecture of the CNN model. There is an input layer, four convolution layers, three max-pooling layers, and a final fully connected layer.

**Figure 3 sensors-21-05456-f003:**
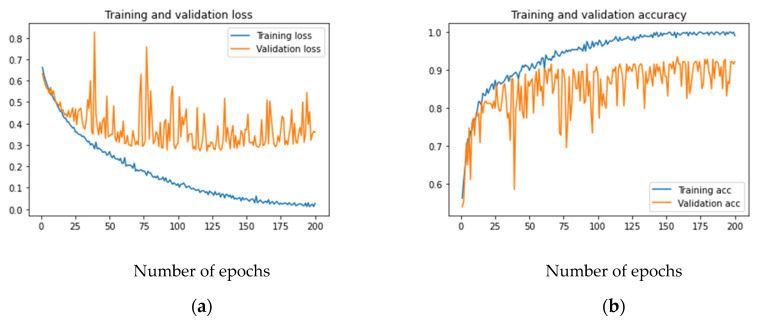
Training and validation loss and accuracy of the baseline CNN model: (**a**) loss; (**b**) accuracy.

**Table 1 sensors-21-05456-t001:** Comparing performance of the baseline and regularized CNN model using various metrics.

CNN Model	Accuracy	Precision	Recall	F1-Score	AUC	Kappa
Baseline	91.15%	92.22%	89.24%	90.71%	91.08%	82.25%
Regularized	98.43%	100%	96.77%	98.36%	98.38%	96.87%

**Table 2 sensors-21-05456-t002:** Result of K-fold cross-validation (K = {3, 5, 10}) accuracy with varying batch size (2^2^ to 2^8^).

	3-Fold		5-Fold		10-Fold		
	Validation	Test	Validation	Test	Validation	Test	Best Run
Samples	256	192	153	192	76	192	
Batch size	μ (σ)		μ (σ)		μ (σ)		
4	0.92 (0.01)	0.96	0.92 (0.01)	0.93	0.95 (0.01)	0.97	0.97
8	0.93 (0.01)	0.94	0.94 (0.02)	0.95	0.94 (0.02)	0.96	0.97
16	0.92 (0.01)	0.94	0.94 (0.01)	0.96	0.94 (0.01)	0.98	0.96
32	0.93 (0.02)	0.95	0.93 (0.03)	0.95	0.94 (0.03)	0.96	0.97
64	0.92 (0.01)	0.95	0.94 (0.01)	0.95	0.94 (0.03)	0.95	1.0
128	0.92 (0.01)	0.90	0.92 (0.02)	0.95	0.95 (0.03)	0.95	0.99
256	0.88 (0.02)	0.89	0.90 (0.03)	0.90	0.90 (0.04)	0.89	0.97

**Table 3 sensors-21-05456-t003:** Comparison of approaches for UCI-ML EEG classification dataset.

Approach	Feature Extractor	Classifiers	Performance
Transfer learning [21]	GLM, Hu moment, LBP + 12 CNN models	KNN, SVM linear/poly/RBF, RF, MLP, and NB Best: SVM RBF	Accuracy: 95.33
			Precision: 95.68
			Recall: 95.00
			F1-score: 95.24
Machine learning [29]	AE, SE, mean, std	SVM cubic/quadratic, KNN, ensemble tree Best: quadratic SVM	Accuracy: 95
			Sensitivity: 95
			AUC: 98
Hybrid Features + EELM [31]	AR, WT, WPD, SE, and class separability	ELM, bagging, boosting Best: LDA + EELM	Accuracy: 91.17
ML + MLP [36]	Min/max, mean, std, power value, Daubechies, coiflets, symlets, and biorthogonal wavelets	SVM, OPF, KNN, NB, MLP Best: NB	Accuracy: 99.6
			Specificity: 99.6
			Sensitivity: 99.6
			PPV: 99.6
MP-CNN [37]	5 MP-CNN models	Best: 19 best channels in CNN with 3 convolution layers and softmax classifier	Accuracy: 100
			Specificity: 100
			Sensitivity: 100
			F1-score: 100
2D-CNN [22]	PCC and 2D spectrograms followed by CNN	CNN with four convolution and pooling layers	Accuracy: 98.13
			Specificity: 97
			Sensitivity: 98
			F1-score: 98
Our approach, CNN	CNN	CNN with 3 convolution layers, dropout, batch normalization, and kernel regularization and softmax classifier on two channels	Accuracy: 98
			Precision: 100
			Recall: 96.8
			F1-score: 98.4
			AUC: 98.4

**Table 4 sensors-21-05456-t004:** A comparison of the baseline and the regularized CNN model architectures.

Baseline CNN Model			Regularized CNN Model		
Layer (Type)	Output Shape	Params	Layer (type)	Output Shape	Params
**Conv1D**	(None, 498, 16)	256	**Conv1D**	(None, 498, 16)	256
Max Pooling 1D	(None, 249, 16)	0	Max Pooling 1D	None, 249, 16)	0
**Conv1D**	(None, 235, 32)	7712	Batch Normal	None, 249, 16)	64
Max Pooling 1D	(None, 117, 32)	0	Dropout	None, 249, 16)	0
**Conv1D**	(None, 103, 64)	30,784	**Conv1D**	(None, 235, 32)	7712
**Conv1D**	(None, 89, 64)	61,504	Max Pooling 1D	(None, 117, 32)	0
Global Max Pooling	(None, 64)	0	Batch Normal	(None, 117, 32)	128
Dense	(None, 1)	65	Dropout	(None, 117, 32)	0
Total params		100,321	**Conv1D**	(None, 103, 64)	30,784
Trainable params		100,321	**Conv1D**	(None, 89, 64)	61,504
Nontrainable params		0	Global Max Pooling	(None, 64)	0
			Batch Normal	(None, 64)	256
			Dropout	(None, 64)	0
			Dense	(None, 1)	65
			Total params		100,769
			Trainable params		100,545
			Nontrainable params		224

## Data Availability

The raw data are available in public domain at [40] as an archive consisting of a number of text files for different channels and subjects. However, it needs to be converted into a format suitable for processing by programming languages like Python. We have made available a single comma-separated values (CSV) file that contains the records for efficient processing in any programming language. The CSV formatted dataset is available at the Kaggle [64], and the code is also available at https://www.kaggle.com/yahamid/eeg-classification-normal-vs-alcoholic-cnn (accessed on 4 August 2021) for interested readers for further experiments.

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
