# Peer review of "Deep Convolutional Neural Network Regularization for Alcoholism Detection Using EEG Signals"

_sensors, 2021, doi:10.3390/s21165456_

Round 1

Reviewer 1 Report

Dear authors,

I present my suggestions and evaluation of your manuscript “Deep Convolutional Neural Network Regularization for Alcoholism Detection Using EEG Signals” below. I consider the topic to be current.

A am really missing the classical structure of scientific paper: introduction -> methods -> results -> discussion -> conclusion. First time I was reading the manuscript I thought that this was a review. I must mention that section 2 has an important role in EEG classification as well, but it can be better set to the manuscript. The discussion should be a separate section and must be improved.  

I have some questions and comments on this manuscript which have been divided into major and minor ones. Please, see below.

Major:

  1. Dataset - The dataset has been mistakenly cited by the last used article. Please revise it. The UCI KDD Archive should be cited as well.
    Row 275: “The dataset for EEG signals was obtained from the UCI-KDD1 online repository that is made 275 available for research [43].”
  2. Which features were chosen for consequent classification?
  3. The introduction should be rewritten. Please try to lead it from a general to a concrete problem or aim that corresponds to the main aims of the manuscript. It can be divided into subsections.
  4. The discussion section must be improved. The comparison with similar studies should be discussed. This section should be separate.
  5. Subsection 5.1 Experimental Setup is a methodology, not a result.
  6. How about preprocessing? You have mentioned a pass-band of [0.02 Hz, 50 Hz]. Was it done by analog filters? Other filtration or artifacts rejection was performed. Data contains the ECG artifact (based on Figure 1). Did You study if this artifact is involved in the whole dataset? Because I think that it can have an influence on classification (based on the training, testing, and validation sets).
  7. Plenty of articles defined using the CNN in the case of EEG classification. You chose to design your own CNN. Why don't you try the already proposed CNN design?
  8. How about the hyperparameters tuning.  Subsection 3.6 Hyperparameters Tuning talks about the tuning in general but how were the hyperparameters tuned in this study?
  9. You mentioned the segmentation in 2s segments. How is it in the context of dividing into training, testing, and validation dataset? Were used segments from each subject for training?
  10. The conclusion is that by the proposed methodology you achieved 98 % accuracy. But you already mentioned that it reached the accuracy of 100 % in the study of Bhuvaneshwari et al. 2021 [41] without any further comparison and discussion. So what is the benefit of this study?

Minor:

  1. Every graph in the manuscript should be described by the variable and unit. Please revise the manuscript and add this information to the figures.
  2. I misunderstood the way CNN was used. Was it used for feature selection? Or whole segments are input for CNN? Or for both of these? I think this should be described more properly.
  3. The database is publicly available, please rewrite the information about supplementary materials. 
    Row 561: “Supplementary Materials: The data and code will be made available after publication.”
  4. There are typographical mistakes in the manuscript (like EGG). Please, try to clear the manuscript.
  5. Why do you choose to avoid the normalization before CNN? Is it typical for EEG classification?

Reviewer 2 Report

In this work, the authors apply convolutional neural networks (CNN) on raw electroencephalogram (EEG) data with the aim to classify the corresponding activity as that of a normal person or an alcoholic person. The obtained results demonstrate that it is possible to achieve 98% average accuracy by optimizing a baseline CNN model for automatic feature selection in the temporal domain of signals and outperforming its results in a range of performance evaluation metrics on the UCI-KDD EGG dataset. 

In my opinion, the study could have important implications, in fact, as also stated by authors, processing techniques have been applied traditionally for EEG signal processing. The approach proposed by the authors seems very efficient in comparison with many others that can be found in literature and extends what can be achieved by using unsupervised statistical approaches such as Principal Component Analysis.

The work is well written, the results are sound, and no serious faults occur throughout the manuscript. In addition, the step-wise description for training the network and applying the model can be used also for a wide range of applications of CNN. This is another strength of the work: I believe that this study can pave the way allowing the discrimination between (and/or increasing the sensitivity of) the signals acquired by different kinds of electrodes.
For the exposed reasons, I believe the manuscript can be accepted for publication on Sensors as it is.

Reviewer 3 Report

This paper is a binary classification of alcohol dependence EEG. However, the machine learning used for classification is basic, and ingenuity such as batch regularization is not a new technique in the field of CNN.

Giri, EP, Fanany, MI, Arymurthy, AM, & Wijaya, SK (2016, October). Ischemic stroke identification based on EEG and EOG using ID convolutional neural network and batch normalization. In 2016 International Conference on Advanced Computer Science and Information Systems (ICACSIS) (pp. 484-491). IEEE.

The following treatises use CNN.

Rahman, S., Sharma, T., & Mahmud, M. (2020, September). Improving alcoholism diagnosis: comparing instance-based classifiers against neural networks for classifying EEG signal. In International Conference on Brain Informatics (pp. 239-250) ). Springer, Cham.

Srabonee, JF, Peya, ZJ, Akhand, MAH, & Siddique, N. (2021). Alcoholism Detection from 2D Transformed EEG Signal. In Proceedings of International Joint Conference on Advances in Computational Intelligence (pp. 297-308). Springer, Singapore.

Silva, FH, Medeiros, AG, Ohata, EF, & Reboucas Filho, PP (2020, July). Classification of Electroencephalogram Signals for Detecting Predisposition to Alcoholism using Computer Vision and Transfer Learning. In 2020 IEEE 33rd International Symposium on Computer-Based Medical Systems (CBMS) (pp. 126-131). IEEE.

In addition, I am comparing it to a baseline model, but the baseline accuracy is low and it is not suitable for comparison. There are treatises such as [39] and [40] that can be classified with 100% probability, so you should compare them. (It is necessary to show the usefulness of this method in detail based on Table 1 and the above paper). The purpose is to investigate simple machine learning, but what is the reason? Real-timeness, memory, learning speed, number of datasets needed for learning, and other reasons? These must also be properly compared with models such as previous papers [39] [40].

Round 2

Reviewer 1 Report

Dear authors,

I am satisfied with the amendments you have made. I think that your work is clear and can be reproduced now.

Reviewer 3 Report

I have confirmed that the author has modified the treatise appropriately.